

# Cortical modulation of pupillary function: systematic review

Costanza Peinkhofer[1,2,*], Gitte M. Knudsen[1,3,4], Rita Moretti[2,5] and Daniel Kondziella[1,4,6,*]

[1] Department of Neurology, Rigshospitalet, Copenhagen University Hospital, Copenhagen, Denmark
[2] Medical Faculty, University of Trieste, Trieste, Italy
[3] Neurobiology Research Unit, Rigshospitalet, Copenhagen University Hospital, Copenhagen, Denmark
[4] Faculty of Health and Medical Science, University of Copenhagen, Copenhagen, Denmark
[5] Department of Medical, Surgical and Health Sciences, Neurological Unit, Trieste University Hospital, Cattinara, Trieste, Italy
[6] Department of Neuroscience, Norwegian University of Technology and Science, Trondheim, Norway
* These authors contributed equally to this work.

Corresponding author
Daniel Kondziella,
daniel_kondziella@yahoo.com

## ABSTRACT

**Background.** The pupillary light reflex is the main mechanism that regulates the pupillary diameter; it is controlled by the autonomic system and mediated by subcortical pathways. In addition, cognitive and emotional processes influence pupillary function due to input from cortical innervation, but the exact circuits remain poorly understood. We performed a systematic review to evaluate the mechanisms behind pupillary changes associated with cognitive efforts and processing of emotions and to investigate the cerebral areas involved in cortical modulation of the pupillary light reflex.

**Methodology.** We searched multiple databases until November 2018 for studies on cortical modulation of pupillary function in humans and non-human primates. Of 8,809 papers screened, 258 studies were included.

**Results.** Most investigators focused on pupillary dilatation and/or constriction as an index of cognitive and emotional processing, evaluating how changes in pupillary diameter reflect levels of attention and arousal. Only few tried to correlate specific cerebral areas to pupillary changes, using either cortical activation models (employing micro-stimulation of cortical structures in non-human primates) or cortical lesion models (e.g., investigating patients with stroke and damage to salient cortical and/or subcortical areas). Results suggest the involvement of several cortical regions, including the insular cortex (Brodmann areas 13 and 16), the frontal eye field (Brodmann area 8) and the prefrontal cortex (Brodmann areas 11 and 25), and of subcortical structures such as the locus coeruleus and the superior colliculus.

**Conclusions.** Pupillary dilatation occurs with many kinds of mental or emotional processes, following sympathetic activation or parasympathetic inhibition. Conversely, pupillary constriction may occur with anticipation of a bright stimulus (even in its absence) and relies on a parasympathetic activation. All these reactions are controlled by subcortical and cortical structures that are directly or indirectly connected to the brainstem pupillary innervation system.

## INTRODUCTION

The pupillary light reflex is a polysynaptic reflex that requires cranial nerves II and III, as well as central brainstem connections (*Kawasaki, 1999*). Light falling into one eye stimulates retinal photoreceptors, bipolar cells and subsequently retinal ganglion cells whose axons form the optic nerve. Some of these axons terminate in the pretectum of the mesencephalon and pretectal neurons project further to the Edinger-Westphal nuclei. Then, preganglionic parasympathetic axons synapse with ciliary ganglion neurons which in turn send postganglionic axons to innervate the pupillary constrictor muscles of both eyes. Conversely, pupillary dilatation relies on the sympathetic system which consists of pre-ganglionic fibers projecting from the hypothalamus to the superior cervical ganglion and post-ganglionic fibers projecting to the iris dilator muscles, via ciliary nerves (*Kawasaki, 1999*).

In addition to brainstem pathways, there exists also a cortical component of pupillary innervation. For instance, emotional responses such as surprise and cognitive processes such as decision making, memory recall and mental arithmetic may produce pupillary dilation (*Steinhauer, Condray & Kasparek, 2000*; *Simpson & Hale, 1969*; *De Gee, Knapen & Donner, 2014*). Pupillary function may be assessed as changes in pupillary size relative to resting state diameter or alterations of the light reflex in terms of reflex amplitude and latency (i.e., time from light stimulus to pupillary constriction). Cognitive scientists and psychologists have used measurements of pupillary diameters since the 1960ies to monitor mental processes in healthy volunteers and people with a wide range of neurological and psychiatric disorders, including Alzheimer's disease, autism and anxiety (*Bittner et al., 2014*; *Lim et al., 2016*; *Bakes, Bradshaw & Szabadi, 1990*; *Krach et al., 2015*). Testing of emotional processes usually involves neutral versus emotionally salient stimuli, e.g., pictures of everyday life objects versus pictures evoking sadness, anger or happiness, whereas cognitive processes are investigated with tasks such as arithmetic calculations and memory recall tests (*Steinhauer, Condray & Kasparek, 2000*; *Van Steenbergen & Band, 2013*). In addition, neuroimaging, including computed tomography (CT) and magnetic resonance imaging (MRI), has been used to correlate changes in pupillary functions with cerebral lesions in patients with stroke and other brain disorders (*Peinkhofer et al., 2018*). In the same vein, electrical stimulation of cortical areas such as the frontal eye field (Brodmann area 8) has been investigated to correlate pupillary and cortical function in non-human primates (*Becket Ebitz & Moore, 2017*).

Although pupillary function is of considerable interest to neurologists, ophthalmologists, neuroscientists, physiologists and psychologists, the exact mechanisms of supratentorial modulation of pupillary function remains poorly understood. Previous (unsystematic) reviews have focused mainly on cognitive aspects such as attention but not on pupillary cortical control (*Laeng, Sirois & Gredebäck, 2012*; *Granholm & Steinhauer, 2004*; *Van der Wel & Van Steenbergen, 2018*).

Therefore, in this review we aimed to identify (a) the cortical and subcortical areas and (b) the behavior and cognitive processes that modulate pupillary function in humans and non-human primates.

## METHODS

We performed a systematic review of the literature using a predefined search strategy and phrasing research objectives with the PICO approach (a standardized way of defining research questions, focusing on Patients, Intervention, Comparison, and Outcome) (*Schardt, Adams & Owens, 2007*). The review was registered with PROSPERO registration number CRD42018116653 (https://www.crd.york.ac.uk/prospero/). The review protocol can be accessed from the online File S1.

### Objectives
#### Primary research objectives

PICO 1: In patients with focal cerebral lesions due to e.g., stroke, traumatic brain injury or brain surgery (P), does involvement of salient cortical and subcortical gray matter areas, including but not limited to the prefrontal eye field, insular cortex and thalamus (I), as compared to healthy controls or neurological patients without such lesions (C), lead to changes of pupillary function, i.e., the light reflex or resting state pupillary diameter (O)?

PICO 2: In healthy human subjects (P), do cognitive efforts (e.g., decision making or mental arithmetic) and processing of non-painful emotional stimuli (I), as compared to task negative and emotionally neutral conditions (C), lead to changes of pupillary function, i.e., the light reflex or resting state pupillary diameter (O)?

#### Secondary research objectives

PICO 3: In non-human primates (P), does invasive experimental manipulation (e.g., electrical stimulation) of cortical and subcortical gray matter areas (I), as compared to absence of stimulation (C), lead to changes of pupillary function, i.e., the light reflex or resting state pupillary diameter (O)?

PICO 4: In non-human primates (P), do cognitive efforts such as decision making and processing of non-painful emotional stimuli (I), as compared to task negative and emotionally neutral conditions (C), lead to changes of pupillary function, i.e., the light reflex or resting state pupillary diameter (O)?

### Eligibility criteria
#### Types of studies

We evaluated all cross-sectional or longitudinal, retrospective or prospective, observational, clinical and research studies as well as interventional trials, including experimental animal work on non-human primates, reporting on pupillary function as related to modulation by cortical and subcortical lesions or stimulations, as well as modulation by cognitive and emotional processes. We excluded reviews and meta-analysis, non-original studies and studies with $n = \leq 15$ human subjects.

#### Participants

All patients aged $\geq 18$ years with ischemic or hemorrhagic stroke, brain trauma and/or brain surgery as well as healthy subjects studied in order to correlate pupillary function with focal lesions and/or to specific cognitive or emotional cerebral processing related to experimental invasive or non-invasive stimulation were included. For secondary research

questions we included non-human primates with or without cerebral lesions studied to correlate pupillary function with cerebral cortical and/or subcortical gray matter areas and with specific cognitive or emotional cerebral processing related to experimental invasive or non-invasive stimulation. For exclusion criteria, the reader is referred to the protocol review (File S1).

### Outcome measures

The main outcome measure was a change in pupillary function, i.e., either a variation of the pupillary diameter or a difference in the light reflex (e.g., a longer latency period), compared to a baseline value or a control group.

### Index tests and interventions

The index tests comprised neuroimaging (CT, MRI including functional MRI, PET, SPECT), post-mortem examination revealing the extent of brain lesions, quantitative pupillometry (Eye Link 1000 and similar devices) and visual inspection of pupillary function. Concerning interventions, we included all studies with invasive procedures such as electrical cortical and/or subcortical stimulation or induced cerebral lesions as well as non-invasive interventions such as cognitive and emotional tasks or sensorial stimulation of healthy humans, humans with specific cerebral lesions (see above) and non-human primates.

## Search methods for identification of studies
### Electronic literature search strategy

We searched MEDLINE (PubMed), EMBASE and Scopus for relevant literature from January 1st, 1960 to November 15th, 2018. As a search strategy, we used both free text-words (TW) and controlled terms obtained with medical subject headings (MeSH). For search strategy and search terms refer to review protocol (File S1). Reference lists were manually screened for further relevant articles.

## Data collection and analysis
### Selection of studies, data extraction and management

Titles and abstracts were first reviewed. Eligible studies were assessed on the basis of their full text and referenced using Mendeley Software (https://www.mendeley.com). Data were extracted by the first author and checked by the senior author. Preferred Reported Items for Systematic reviews and Meta-analyses (PRISMA) guidelines were followed (*Liberati et al., 2009*) (see File S2).

## RESULTS

We screened 8809 papers in the primary search; three additional publications were manually added. After the exclusion of duplicates, studies with different topic and subjects below 18 years of age, 856 citations were screened for eligibility criteria on an abstract basis. Three-hundred and fifty-five articles were analyzed with a full text review, and 258 studies were included for the final analysis. Figure 1 provides a flowchart of the literature search.

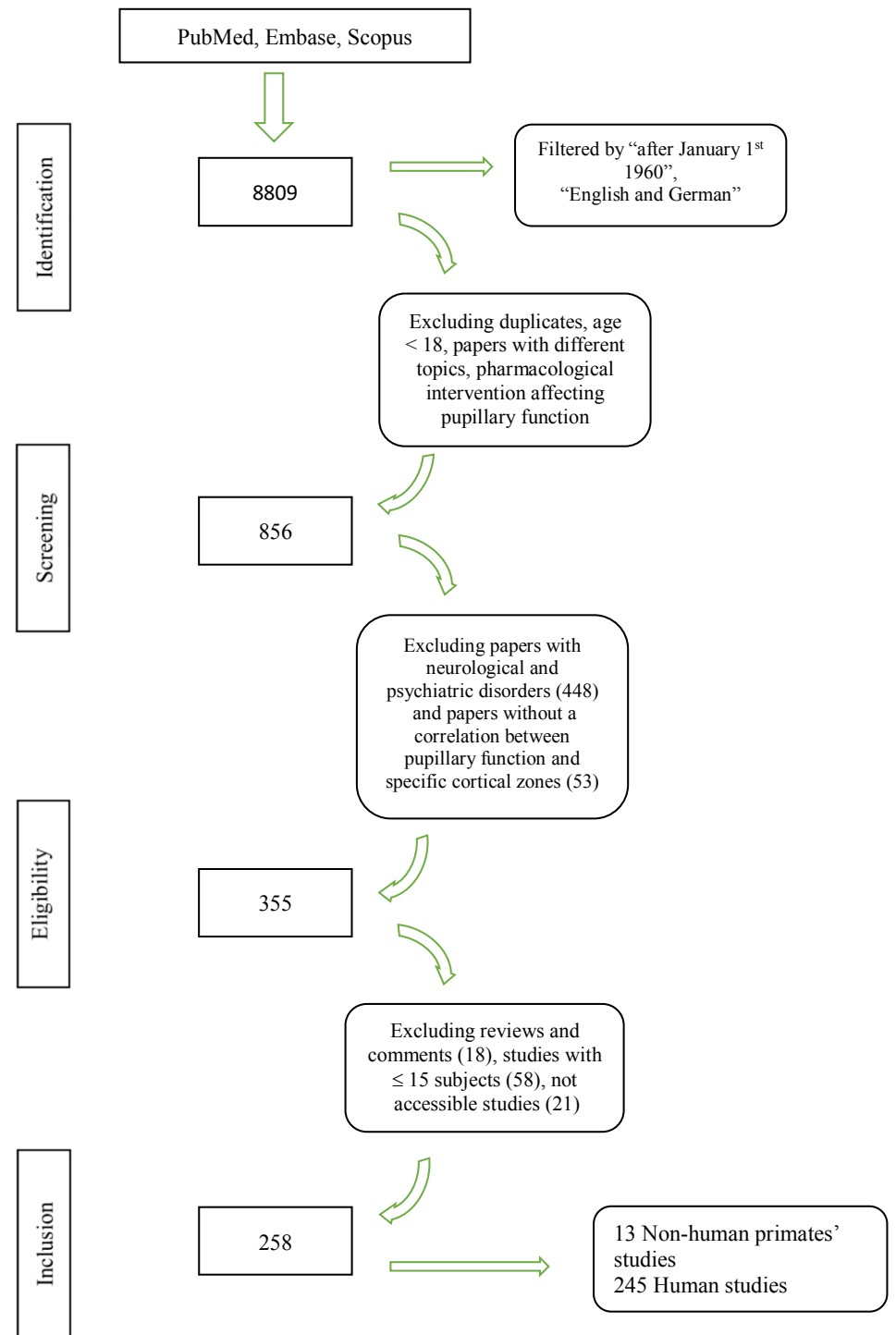

**Figure 1** **Flowchart of the literature search.** Flowchart showing the literature search and the study selection process.

## PICO 1: Pupillary changes associated with cortical lesions in humans

Cerebral areas that may modulate the pupillary light reflex were examined in three studies involving patients with cerebrovascular lesions. One study assessed pupillary dilatation as an index of arousal and reward processing during an oculomotor capture task (*Manohar et al., 2016*), revealing diminished pupillary dilatation in patients with chronic ventromedial prefrontal damage (Brodmann areas 11 and 25) due to subarachnoid hemorrhage as compared to healthy controls. Another, retrospective study of patients with cerebrovascular lesions (*Herman, 1975*), showed persistent anisocoria associated with lesions involving the right or left middle cerebral artery (MCA) territory in the absence of oculomotor nerve compression, but neuroimaging was not available and study results should be cautiously interpreted. In contrast, ischemic stroke lesions were verified using CT in a recent, prospective study, in which investigators assessed how anterior circulation strokes involving the prefrontal eye field (Brodmann area 8) and/or the insular cortex (Brodmann areas 13 and 16) affected pupillary function. Patients with strategic infarcts in either of these areas showed subtle differences during the dilatation phase of the pupillary light reflex, but not patients with infarcts in other cerebral regions or neurologically normal controls (*Peinkhofer et al., 2018*).

## PICO 2: Pupillary changes associated with cognitive and emotional activity in humans

Most of the papers ($n = 242$) referred to changes in pupillary diameter during cognitive and/or emotional processes in humans. One hundred eighty-one (75%) assessed pupillary diameter as an index of mental effort during different cognitive activities. Sixty-one studies (25%) focused on the relationship between emotional arousal and pupillary reaction (Table 1).

### *Pupillary constriction: cognition and emotions*

Constriction was observed in response to specific attentional tasks, where subjects had to focus on luminous stimuli such as bright surfaces (*Turi, Burr & Binda, 2018*; *Mathôt et al., 2014*), illusory or mental images of brightness (*Laeng & Endestad, 2012*; *Laeng & Sulutvedt, 2014*) and pictures of the sun (*Sperandio, Bond & Binda, 2018*; *Naber & Nakayama, 2013*) as opposed to darker stimuli or scattered pictures of the sun (*Sperandio, Bond & Binda, 2018*; *Naber & Nakayama, 2013*). Following the same concept, constriction was also recorded for visually or auditory words conveying luminance (e.g., "lamp") compared to words conveying darkness (e.g.,"night") (*Mathôt, Grainger & Strijkers, 2017*).

A smaller pupillary diameter was also considered an index of novelty during memory formation (i.e., pupillary constriction associated with remembered words) and memory retrieval (i.e., pupillary constriction with forgotten words) (*Naber et al., 2013*). Pupillary constriction may also occur with certain affective responses activating the parasympathetic system such as disgust (*Ayzenberg, Hickey & Lourenco, 2018*).

### *Pupillary dilatation: cognition*

Several studies recorded pupillary dilatation with memory tests, revealing how a change in diameter is related to memory retrieval. Pupillary dilatation occurred during testing

**Table 1  Human studies of the influence of cognitive and emotional processes on pupillary function.** Every study is categorized depending on the specific task required and/or type of stimuli used (*first column on the left*) and on the observed pupillary response (*central and right column*).

| | Pupillary Dilation | Pupillary responses other than dilation |
|---|---|---|
| ***COGNITION*** | | |
| Memory | *Brocher & Graf (2017)*, *Brocher & Graf (2016)*, *Gomes, Montaldi & Mayes (2015)*, *Heaver & Hutton (2011)*, *Johansson et al. (2018)*, *Kafkas & Montaldi (2011)*, *Kafkas & Montaldi (2015)*, *Kafkas & Montaldi (2012)*, *Mill, O'Connor & Dobbins (2016)*, *Montefinese et al. (2013)*, *Otero, Weekes & Hutton (2011)*, *Bradley & Lang (2015)*, *Herweg, Sommer & Bunzeck (2017)*, *Võ et al. (2008)*, *Weiss et al. (2016)*, *Sher (1971)*, *Starc, Anticevic & Repovš (2017)*, *Unsworth & Robison (2018b)*, *Johnson (1971)*, *Klingner, Tversky & Hanrahan (2011)*, *Taylor (1981)*, *Van Gerven et al. (2004)*, *Wong & Epps (2016)*, *Tsukahara, Harrison & Engle (2016)*, *Bijleveld (2018)*, *Cabestrero, Crespo & Quirós (2009)*, *Granholm et al. (1996)*, *Magliero (1983)*, *Piquado, Isaacowitz & Wingfield (2010)*, *Van Rijn et al. (2012)*, *Morey (2018)*, *Robison & Unsworth (2018)*, *Boyer et al. (2018)* and *Bergt et al. (2018)* | Pupillary constriction (*Naber et al., 2013*) |
| Attention including orienting reflex | *Liao et al. (2016)*, *Marois et al. (2018)*, *Steiner & Barry (2011)*, *Stelmack & Siddle (1982)*, *Franklin et al. (2013)*, *Hopstaken et al. (2016)*, *Huijser, Van Vugt & Taatgen (2018)*, *Smallwood et al. (2011)*, *Unsworth & Robison (2016)*, *Brink Van Den, Murphy & Nieuwenhuis (2016)*, *Gouraud, Delorme & Berberian (2018a)*, *Unsworth & Robison (2018a)*, *Gouraud, Delorme & Berberian (2018b)*, *Kang, Huffer & Wheatley (2014)*, *Kang & Wheatley (2017)*, *Wierda et al. (2012)*, *Willems, Herdzin & Martens (2015)*, *Geva et al. (2013)*, *Alnaes et al. (2014)*, *Brocher et al. (2018)*, *Chatham et al. (2012)*, *Hosseini et al. (2017)*, *Koenig, Uengoer & Lachnit (2018)*, *McCloy et al. (2017)*, *Nunnally et al. (1967)*, *Tylén et al. (2012)*, *Unsworth, Robison & Miller (2018)*, *Wahn et al. (2016)*, *Yellin, Berkovich-Ohana & Malach (2015)*, *Chiew & Braver (2013)*, *Wykowska et al. (2013)*, *Sulutvedt, Mannix & Laeng (2018)*, *Quirins et al. (2018)* and *Campbell, Toth & Brady (2018)* | Pupillary constriction (*Turi, Burr & Binda, 2018*; *Mathôt et al., 2014*; *Laeng & Endestad, 2012*; *Laeng & Sulutvedt, 2014*; *Sperandio, Bond & Binda, 2018*; *Naber & Nakayama, 2013*) |
| Language processing and learning | *Mathôt, Grainger & Strijkers (2017)*, *Reinhard, Lachnit & König (2006)*, *Koelewijn et al. (2012b)*, *Winn, Edwards & Litovsky (2015)*, *Zekveld, Kramer & Festen (2010)*, *Zekveld, Festen & Kramer (2013)*, *Zekveld et al. (2014a)*, *Zekveld & Kramer (2014)*, *Zekveld et al. (2014b)*, *Kuchinke et al. (2007)*, *Papesh & Goldinger (2012)*, *Schmidtke (2014)*, *Colman & Paivio (1970)*, *Engelhardt, Ferreira & Patsenko (2010)*, *Paivio & Simpson (1966)*, *Simpson & Paivio (1968)*, *Ben-Nun (1986)*, *Schluroff et al. (1986)*, *Tromp, Hagoort & Meyer (2016)*, *Borghini & Hazan (2018)*, *Iacozza, Costa & Duñabeitia (2017)*, *Foroughi, Sibley & Coyne (2017)*, *Shalev et al. (2018)*, *Reinhard & Lachnit (2002)*, *Van Der Meer et al. (2003)*, *Kahya et al. (2018)*, *Ariel & Castel (2014)*, *Bayer, Sommer & Schacht (2011)*, *Beatty & Wagoner (1978)*, *Carver (1971)*, *Causse et al. (2010)*, *Demberg & Sayeed (2016)*, *Just & Carpenter (1993)*, *Koelewijn et al. (2012a)*, *Fernández et al. (2016)*, *Hyona, Tommola & Alaja (1995)*, *Hoffing & Seitz (2015)*, *Koelewijn et al. (2015)*, *Kramer et al. (2013)*, *Kuipers & Thierry (2011)*, *Laeng et al. (2011)*, *Lobben & Bochynska (2018)*, *Ojha, Indurkhya & Lee (2017)*, *Metalis et al. (1980)*, *Scheepers et al. (2013)*, *Sevilla, Maldonado & Shalóm (2014)*, *Zellin et al. (2011)*, *White & French (2017)* and *Zekveld et al. (2018)* | Pupillary constriction (*Mathôt, Grainger & Strijkers, 2017*) |

**Table 1** (*continued*)

| | Pupillary Dilation | Pupillary responses other than dilation |
|---|---|---|
| Mental arithmetic | *Steinhauer, Condray & Kasparek (2000), Klingner, Tversky & Hanrahan (2011), Bradshaw (1967), Chen & Epps (2014), Marquart & De Winter (2015), Szulewski, Roth & Howes (2015), Szulewski et al. (2017), Steinhauer et al. (2004)* and *Annerer-Walcher, Körner & Benedek (2018)* | Attenuated light reflex (*Steinhauer, Condray & Kasparek, 2000*) |
| Decision making including uncertainty | *De Gee, Knapen & Donner (2014), Mill, O'Connor & Dobbins (2016), Szulewski, Roth & Howes (2015), Szulewski et al. (2017), Verney, Granholm & Dionisio (2001), Verney, Granholm & Marshall (2004), Wolff et al. (2015), Reilly et al. (2018), Trani & Verhaeghen (2018), Stojmenova & Sodnik (2018), Jepma & Nieuwenhuis (2011), Katidioti, Borst & Taatgen (2014), Oliva & Anikin (2018), Berthold & Slowiaczek (1975), Schneider et al. (2018), Lempert, Chen & Fleming (2015), Lin et al. (2017), Satterthwaite et al. (2007), Brunyé & Gardony (2017), Geng et al. (2015), Einhäuser, Koch & Carter (2010), Fehrenbacher & Djamasbi (2017), Prehn, Heekeren & Van der Meer (2011), Porter et al. (2010), Rigato, Rieger & Romei (2016), Schlemmer et al. (2005)* and *Mitra, McNeal & Bondell (2017)* | |

Various:

| | Pupillary Dilation | |
|---|---|---|
| -Deception | *Bradley & Janisse (1981), Dionisio et al. (2001), Seymour, Baker & Gaunt (2013)* and *Webb et al. (2009)* | |
| -Time and preparatory activity | *Akdoğan, Balci & Van Rijn (2016), Landgraf, Raisig & Van Der Meer (2012), Nowack, Milfont & Van der Meer (2013), Nuthmann & Van der Meer (2005), Irons, Jeon & Leber (2017), Kahneman, Onuska & Wolman (1968), Moresi et al. (2008), Ribeiro & Castelo-Branco (2019)* and *Massar et al. (2018)* | |
| -Conflict processing | *Van Steenbergen & Band (2013), D'Ascenzo et al. (2016)* and *Diede & Bugg (2017)* | |
| -Error | *Braem et al. (2015), Harsay et al. (2017), Raisig et al. (2007)* and *Raisig et al. (2010)* | |
| -Mental workload | *Juris & Velden (1977), Reiner & Gelfeld (2014)* and *Wright, Boot & Morgan (2013)* | |

***EMOTION/AROUSAL***

Preference for

| | Pupillary Dilation | |
|---|---|---|
| Faces | *Allard, Wadlinger & Isaacowitz (2010), Blackburn & Schirillo (2012), Bradley et al. (2008), Chiesa et al. (2015), Goldinger, He & Papesh (2009), Kret et al. (2013), Lichtenstein-Vidne et al. (2017), Porter, Hood & Troscianko (2006), Schrammel et al. (2009), Vanderhasselt et al. (2018), Wu, Laeng & Magnussen (2012), Yrttiaho et al. (2017), Kret (2017)* and *Hammerschmidt et al. (2018)* | |
| Political candidates | *Barlow (1969)* | |
| Visual arts | *Elschner, Hübner & Dambacher (2017), Hayes, Muday & Schirillo (2013), Kuchinke et al. (2009), Powell & Schirillo (2011), Schirillo (2014)* and *Alvarez et al. (2015)* | |
| Alcoholic beverages | *Beall (1977)* | |

**Table 1** (*continued*)

| | Pupillary Dilation | Pupillary responses other than dilation |
|---|---|---|
| Neutral versus emotional stimulus | *Bradley, Sapigao & Lang (2017)*, *Henderson, Bradley & Lang (2017)*, *Iijima et al. (2014)*, *Chiew & Braver (2014)*, *Pearlstein et al. (2018)*, *Thoma & Baum (2018)*, *Metalis & Hess (1982)*, *Babiker et al. (2015)*, *Gingras et al. (2015)*, *Laeng et al. (2016)*, *Rosa et al. (2017)*, *Widmann, Schröger & Wetzel (2018)*, *Park & Whang (2018)*, *Leuchs, Schneider & Spoormaker (2018)*, *Wollner, Hammerschmidt & Albrecht (2018)*, *Cohen, Moyal & Henik (2015)*, *Snowden et al. (2016)*, *Urry et al. (2009)*, *Vanderhasselt et al. (2014)*, *Bebko et al. (2011)*, *Bardeen & Daniel (2017)*, *Stanners et al. (1979)*, *Kinner et al. (2017)*, *Yih et al. (2018)*, *Nunnally et al. (1967)*, *Ferrari et al. (2016)*, *Siegle et al. (2015)*, *Damsma & Van Rijn (2017)*, *Kloosterman et al. (2015)* and *Bayer, Ruthmann & Schacht (2017)* | Pupillary constriction (*Ayzenberg, Hickey & Lourenco, 2018*) Attenuated light reflex (*Henderson, Bradley & Lang, 2014*; *Cohen, Moyal & Henik, 2015*) |
| Olfactory stimulation | *Aguillon-Hernandez et al. (2015)* and *Schneider et al. (2009)* | |
| Sexual arousal | *Metalis & Hess (1982)*, *Dabbs (1997)*, *Hamel (1974)*, *Rieger & Savin-Williams (2012)* and *Attard-johnson, Ciardha & Bindemann (2018)* | |

of short term and working memory, e.g., recognizing previously presented words, pictures, or sounds (*Brocher & Graf, 2017*; *Brocher & Graf, 2016*; *Gomes, Montaldi & Mayes, 2015*; *Heaver & Hutton, 2011*; *Johansson et al., 2018*; *Kafkas & Montaldi, 2011*; *Kafkas & Montaldi, 2015*; *Kafkas & Montaldi, 2012*; *Mill, O'Connor & Dobbins, 2016*; *Montefinese et al., 2013*; *Otero, Weekes & Hutton, 2011*; *Bradley & Lang, 2015*; *Herweg, Sommer & Bunzeck, 2017*; *Võ et al., 2008*; *Weiss et al., 2016*) or digit-recall tasks (*Sher, 1971*; *Starc, Anticevic & Repovš, 2017*; *Unsworth & Robison, 2018b*; *Johnson, 1971*; *Klingner, Tversky & Hanrahan, 2011*; *Taylor, 1981*; *Van Gerven et al., 2004*; *Wong & Epps, 2016*; *Tsukahara, Harrison & Engle, 2016*; *Bijleveld, 2018*). Pupillary dilatation also reflects information storage and mental overload; memorizing more than five items evoked a pupillary dilatation lasting as long as the stimulus itself (*Unsworth & Robison, 2018b*; *Cabestrero, Crespo & Quirós, 2009*; *Reinhard, Lachnit & König, 2006*). Of note, pupillary dilatation, recorded during an encoding-retrieval phase, is associated with activity in the ventral striatum and in the Globus pallidus as revealed by fMRI, suggesting involvement of these areas in memory formation and pupillary function (*Herweg, Sommer & Bunzeck, 2017*).

Another mental process influencing pupillary diameter is attention, i.e., tasks such as reading and focusing on a target elicit pupillary dilatation. Attention related to the orienting reflex, e.g., associated with sudden noise or a bright stimulus, also elicits pupillary dilatation (*Liao et al., 2016*; *Marois et al., 2018*; *Steiner & Barry, 2011*; *Stelmack & Siddle, 1982*). Conversely, smaller pupil sizes are seen with mind-wandering and introspection, and decreasing pupillary diameters reflect distraction and poor task performance (*Franklin et al., 2013*; *Hopstaken et al., 2016*; *Huijser, Van Vugt & Taatgen, 2018*; *Smallwood et al., 2011*; *Unsworth & Robison, 2016*; *Brink Van Den, Murphy & Nieuwenhuis, 2016*; *Gouraud, Delorme & Berberian, 2018a*; *Unsworth & Robison, 2018a*; *Gouraud, Delorme & Berberian, 2018b*). Pupillary changes can thus uncover the level of attention and the amount of mental effort with high temporal resolution (*Kang, Huffer & Wheatley, 2014*; *Kang & Wheatley, 2017*; *Wierda et al., 2012*; *Willems, Herdzin & Martens, 2015*).

Based on the dilatation evoked by hearing and reading sentences, several authors assessed pupillary diameters to categorize language and word processing. Pupils dilate more with poor intelligibility (*Koelewijn et al., 2012b*; *Winn, Edwards & Litovsky, 2015*; *Zekveld, Kramer & Festen, 2010*; *Zekveld, Festen & Kramer, 2013*; *Zekveld et al., 2014a*; *Zekveld & Kramer, 2014*; *Zekveld et al., 2014b*) and increased effort for low compared to high frequency words (*Kuchinke et al., 2007*; *Papesh & Goldinger, 2012*; *Schmidtke, 2014*), as well as for abstract compared to concrete words (*Colman & Paivio, 1970*; *Engelhardt, Ferreira & Patsenko, 2010*; *Paivio & Simpson, 1966*; *Simpson & Paivio, 1968*). Thus, pupillary dilatation reflects the amount of processing required for understanding of complex or ambiguous sentences (*Ben-Nun, 1986*; *Schluroff et al., 1986*; *Tromp, Hagoort & Meyer, 2016*) and allow to explore differences between native and non-native speakers (*Schmidtke, 2014*; *Borghini & Hazan, 2018*; *Iacozza, Costa & Duñabeitia, 2017*).

Measuring the effectiveness of learning may also be monitored through pupillary dilatation. Learning processes such as Pavlovian, associative learning or categorization are characterized by large pupils initially, when the cognitive load is big, and by smaller diameters when the task or item is being learned (*Reinhard, Lachnit & König, 2006*; *Foroughi, Sibley & Coyne, 2017*; *Shalev et al., 2018*; *Reinhard & Lachnit, 2002*; *Van Der Meer et al., 2003*; *Kahya et al., 2018*). Pupils also dilate in response to mental arithmetic (*Steinhauer, Condray & Kasparek, 2000*; *Bradshaw, 1967*; *Chen & Epps, 2014*; *Marquart & De Winter, 2015*; *Szulewski, Roth & Howes, 2015*; *Szulewski et al., 2017*; *Steinhauer et al., 2004*; *Annerer-Walcher, Körner & Benedek, 2018*), decision-making and visual backward masking tasks (*Verney, Granholm & Dionisio, 2001*; *Verney, Granholm & Marshall, 2004*; *Schneider et al., 2018*; *Wolff et al., 2015*; *Reilly et al., 2018*; *Trani & Verhaeghen, 2018*; *Stojmenova & Sodnik, 2018*; *Jepma & Nieuwenhuis, 2011*; *Katidioti, Borst & Taatgen, 2014*; *Oliva & Anikin, 2018*; *Berthold & Slowiaczek, 1975*) and they can reveal the degree of certainty during any selection process, i.e., the more undecided one is, the greater the pupillary diameter (*Lempert, Chen & Fleming, 2015*; *Lin et al., 2017*; *Satterthwaite et al., 2007*; *Brunyé & Gardony, 2017*; *Geng et al., 2015*).

### Pupillary dilatation: emotions

Stimuli causing emotional arousal can be revealed by changes in pupillary diameter. For instance, pupillary dilatation reflects preference for political candidates (*Barlow, 1969*), alcoholic beverages (*Beall, 1977*) and visual arts (e.g., Rembrandt's paintings) (*Elschner, Hübner & Dambacher, 2017*; *Hayes, Muday & Schirillo, 2013*; *Kuchinke et al., 2009*; *Powell & Schirillo, 2011*; *Schirillo, 2014*; *Alvarez et al., 2015*) allowing to predict people's tastes. Images of human faces elicit a pupillary reaction as well: Angry or fearful facial expressions and images of females increase pupil sizes, in contrast to happy faces and males' images (*Allard, Wadlinger & Isaacowitz, 2010*; *Blackburn & Schirillo, 2012*; *Bradley et al., 2008*; *Chiesa et al., 2015*; *Goldinger, He & Papesh, 2009*; *Kret et al., 2013*; *Lichtenstein-Vidne et al., 2017*; *Porter, Hood & Troscianko, 2006*; *Schrammel et al., 2009*; *Vanderhasselt et al., 2018*; *Wu, Laeng & Magnussen, 2012*; *Yrttiaho et al., 2017*; *Kret, 2017*; *Hammerschmidt et al., 2018*). Negative images showing violence, distress and threat but also positive ones depicting happiness elicited a dilatation as opposed to neutral everyday images (*Henderson,*

*Bradley & Lang, 2014*; *Bradley, Sapigao & Lang, 2017*; *Henderson, Bradley & Lang, 2017*; *Iijima et al., 2014*; *Chiew & Braver, 2014*; *Pearlstein et al., 2018*; *Thoma & Baum, 2018*). Pupillary dilatation may also signal the perception of odors (*Aguillon-Hernandez et al., 2015*; *Schneider et al., 2009*) and sexual arousal (*Metalis & Hess, 1982*; *Dabbs, 1997*; *Hamel, 1974*; *Rieger & Savin-Williams, 2012*; *Attard-johnson, Ciardha & Bindemann, 2018*); salient odors or visual or auditory sexual stimuli lead to pupillary dilatation. Pupillary dilatation results also from pleasant sounds and melodies. Known music tracks enhance pupillary diameters but not unknown and less salient melodies (*Babiker et al., 2015*; *Gingras et al., 2015*; *Laeng et al., 2016*; *Rosa et al., 2017*; *Widmann, Schröger & Wetzel, 2018*; *Park & Whang, 2018*; *Leuchs, Schneider & Spoormaker, 2018*; *Wollner, Hammerschmidt & Albrecht, 2018*). Finally, measures of pupillary diameter may also reveal active mental efforts associated with coping strategies such as reappraisal or suppression of negative emotions (*Cohen, Moyal & Henik, 2015*; *Snowden et al., 2016*; *Urry et al., 2009*; *Vanderhasselt et al., 2014*; *Bebko et al., 2011*; *Bardeen & Daniel, 2017*; *Stanners et al., 1979*; *Kinner et al., 2017*; *Yih et al., 2018*). Neuroimaging studies involving fMRI show that at least some of these emotional conditions leading to pupillary dilatation are associated with increased activation of the amygdala, the ventro-medial prefrontal cortex (Brodmann areas 11 and 25), the lateral occipital complex (*Kuniecki et al., 2018*) and the dorsolateral prefrontal cortex (Brodmann areas 9 and 46) (*Vanderhasselt et al., 2014*).

## PICO 3: Pupillary changes associated with cortical stimulation and lesions in non-human primates

Pupillary dilatation occurs in non-human primates in response to electrical stimulation of the frontal eye field (Brodmann area 8) during passive viewing tasks (*Lehmann & Corneil, 2016*) ("probe in, probe out" conditions (*Becket Ebitz & Moore, 2017*)), and of the superior colliculus (*Wang et al., 2012*; *Joshi et al., 2016*) during passive fixation tasks. One study compared non-human primates with amygdala lesions to healthy controls during a free viewing task; pupillary dilation was similar in both groups, but the pupillary light reflex was diminished in the lesion group (*Dal Monte et al., 2015*) (Table 2).

## PICO 4: Pupillary changes associated with cognitive and emotional activity in non-human primates

As in humans, cognitive processes lead to pupillary dilatation in rhesus macaques. Changes in pupil diameters occur in non-human primates during different tasks such as button pushing (*Iriki et al., 1996*), visual orientation (*Hampson, Opris & Deadwyler, 2010*; *Ebitz, Pearson & Platt, 2014*), recognition and memory (*Montefusco-Siegmund, Leonard & Hoffman, 2017*) or sensorial stimulation (e.g., auditory or electrodermal) (*Joshi et al., 2016*; *Iriki et al., 1996*). Some investigators correlated pupillary function with specific cortical or subcortical areas, recording neuronal firing through implanted electrodes. Neural activity during pupil dilatation was noted in the frontal cortex (Brodmann area 8) (*Hampson, Opris & Deadwyler, 2010*) and both anterior and posterior cingulate cortex (Brodmann areas 23, 24, 31, 32) (*Joshi et al., 2016*; *Ebitz et al., 2015*), as well as in key brainstem structures such as locus coeruleus and the inferior and superior colliculi (*Joshi et al., 2016*) (Table 3).

**Table 2  Non-human primate studies on the relationship of pupillary function with specific cortical/subcortical structures.** List of studies investigating if micro stimulation of some cerebral areas, through previously implanted electrodes, resulted in pupillary changes in diameter.

| Source | Species | Pupillary assessment | Stimulated areas | Task | Pupillary dilation | Pupillary responses other than dilation |
|---|---|---|---|---|---|---|
| Becket Ebitz & Moore (2017) | Rhesus Macaque (n = 2) | Eyelink 1000 (SR Research) | Frontal Eye Field (Area 8) | Fixation (with distraction) | None | Enhanced pupillary light reflex |
| | | | | Fixation (without distraction) | Yes | |
| Joshi et al. (2016) | Rhesus Macaque (n = 5) | Eyelink 1000 (SR Research) | Locus Coeruleus | None | Yes | None |
| | | | Inferior Colliculus | | | |
| | | | Superior Colliculus | | | |
| Lehmann & Corneil (2016) | Rhesus Macaque (n = 2) | ETL 200 (IScan) | Frontal Eye Field (Area 8) | Fixation | Yes | None |
| Wang et al. (2012) | Rhesus Macaque (n = 2) | Eyelink II (SR Research) | Superior Colliculus | Fixation | Yes | None |
| Jampel (1960) | Rhesus Macaque (n = 9) | Visual inspection | Frontal Cortex (Area 8-9-10) | None | Yes | Pupillary constriction and accomodation |
| | | | Occipital Cortex (Area 18-19-22) | | | |
| Dal Monte et al. (2015) | Rhesus Macaque (n = 8) | Arrington View Point | [a]Amygdala lesions | Free viewing | Yes | Reduction of pupillary light reflex |

**Notes.**
[a]Comparison between monkeys with amygdala lesions and healthy controls.

## DISCUSSION

This systematic review reveals that pupils do not only dilate and constrict in response to light, but a large number of cognitive and emotional processes affects pupillary function and leads to pupillary dilatation and, less often, to constriction (Table 1). Pupil diameter may serve as an index of brain activity, reflecting mental efforts (or lack of efforts). Thus, our pupils dilate, when we are focused in contrast to when we let our minds wander; they dilate when we are dishonest and lying; when we enjoy or dislike what we are seeing; and when we are engaged in learning and processing of information.

### Pupillary dilatation

The most commonly observed response following emotional or cognitive tasks is pupillary dilatation. In humans, as well as in non-human primates, this is due to sympathetic

**Table 3  Non-human primate studies on the relationship of cognitive and emotional processes with pupillary function and activation of cortical/subcortical areas.** Characteristics of studies investigating which tasks and/or sensorial stimulus evoked a pupillary response and which cerebral areas were simultaneously activated.

| Source | Species | Pupillary assessment | Cortical and subcortical Recorded activity | Cognitive task | Sensory stimulus | Pupillary dilation | Pupillary responses other than dilation |
|---|---|---|---|---|---|---|---|
| Hampson, Opris & Deadwyler (2010) | Rhesus Macaque (n = 4) | EyeLink 1000 (SR Research) | Frontal Cortex (Area 8) | Visual Delayed Match to Sample | N/A | Yes | None |
| Iriki et al. (1996) | Japanese Macaque (n = 2) | MOS camera under infrared illumination | Somatosensory Cortex (Area 3, Postcentral Gyrus, finger hand region) | Button Pushing | N/A | Yes | None |
| | | | | N/A | Passive Skin Stimulation | No | None |
| Joshi et al. (2016) | Rhesus Macaque (n = 5) | EyeLink 1000 (SR Research) | Locus Coeruleus Inferior and Superior Colliculus, Anterior and Posterior Cingulate Cortex (Areas 32, 23 and 31) | [a] | N/A | Yes | Oscillations |
| | | | | | Startling Tone | Yes | None |
| | | | | Visual Search and Detection | N/A | Yes | None |
| Montefusco-Siegmund, Leonard & Hoffman (2017) | Rhesus Macaque (n = 2) | iViewX Hi-Speed (SBI) | Hippocampus | N/A | Visual presentation of natural scenes | Yes | None |
| Suzuki, Kunimatsu & Tanaka (2016) | Japanese Macaque (n = 3) | iRecHS2 (AIST) | N/A | Time production/ Memory Task | N/A | Yes | None |
| Ebitz et al. (2015) | Rhesus Macaque (n = 2) | EyeLink 1000 (SR Research) | Dorsal Anterior Cingulate Cortex (Area 24) | Task Conflict and Error | N/A | N/A | Differences in pupils' baselines |
| Ebitz, Pearson & Platt (2014) | Rhesus Macaque (n = 4) | EyeLink 1000 (SR Research) | N/A | Visual Orienting With Distractors | N/A | N/A | Differences in pupils' baselines |
| Cash-Padgett et al. (2018) | Rhesus Macaque (n = 2) | EyeLink 1000 (SR Research) | Dorsal and Subgenual Anterior Cingulate (Areas 24,33) | Decision making (gambling task) | N/A | Yes | None |

**Notes.**
[a] No cognitive task required, only fixation.

activation or parasympathetic inhibition or a combination of the two (*Steinhauer, Condray & Kasparek, 2000*) and based on unconscious mechanisms. Hence, tasks that require a high amount of attention such as memory retrieval, mental arithmetic or language processing elicit a sympathetic activation. Similarly, emotional sounds and images induce a state of arousal, which involves sympathetic activity leading to pupillary dilatation.

Cerebral structures involved in vigilance, arousal and attention and responsible for changes in pupillary diameter during cognitive and emotional processes include the locus coeruleus (*Joshi et al., 2016*; *Murphy et al., 2014*), the superior colliculus (*Wang et al., 2012*) and multiple regions of the frontal/prefrontal cortex (Brodmann areas 8, 9 and 11 ) (*Becket Ebitz & Moore, 2017*; *Lehmann & Corneil, 2016*) (Fig. 2). Of these, the locus coeruleus seems to be the most influential mediator of the pupillary light reflex. This pontine nucleus is part of the ascending reticular activating system (ARAS) and intimately and reciprocally linked to the orbitofrontal cortex (Brodmann area 11) and the anterior cingulate cortex (Brodmann area 24 and 32) (*Johansson et al., 2018*; *Geva et al., 2013*) which are both fundamental to motivational relevance and target fixation. Evidence from studies of these networks supports the notion that attention and vigilance are related to the regulation of pupillary light reflex. Thus, the locus coeruleus modulates an excitatory connection to the sympathetic network of the pupil (in particular to the intermediate-medial-lateral cell column of the spinal cord) and an inhibitory connection to the parasympathetic pathway (directing to the Edinger Westphal nucleus). Activation of the locus coeruleus leads to increased sympathetic and decreased parasympathetic activity and, consequently, pupil dilatation (*Samuels & Szabadi, 2008*). Two recent studies highlight these aspects. According to *Joshi et al. (2016)*, the locus coeruleus acts together with the inferior and superior colliculi, as well as the anterior and posterior cingulate cortex (Brodmann areas 23, 24, 31, 32) likely in response to increased vigilance and alertness, thereby modifying the pupillary diameter. The second study (*Schneider et al., 2016*), conducted on human beings, confirms this theory and shows that, based on data from resting state magnetic resonance imaging, pupil dilatation is related to an increased activity of the thalamus and frontoparietal regions (Brodmann areas 6, 39, 40), involved in the so-called tonic alert status and vigilance, and to increased metabolism of the visual and sensory-motor regions.

Besides the locus coeruleus, the superior colliculus seems to play a key role in modulating the pupillary light reflex. *Wang et al. (2012)* reported that pupils temporarily dilate after stimulation of the intermediate layer of the superior colliculus in non-human primates. Further, *Mill, O'Connor & Dobbins (2016)* and *Herweg, Sommer & Bunzeck (2017)* suggested that the superior colliculus receives neuronal inputs from temporal, frontal and parietal areas and basal ganglia, especially striatal and pallidal neuronal groups, leading to pupillary dilatation associated with memory tasks.

In addition, different experimental conditions in macaques show that stimulation of the frontal eye field (Brodmann area 8) might modulate the pupillary light reflex (*Becket Ebitz & Moore, 2017*; *Hampson, Opris & Deadwyler, 2010*). For instance, simultaneous micro-stimulations of the frontal eye field and of pretectum structures enhance the activity of frontal eye field neurons with similar spatial tuning and reduce, or

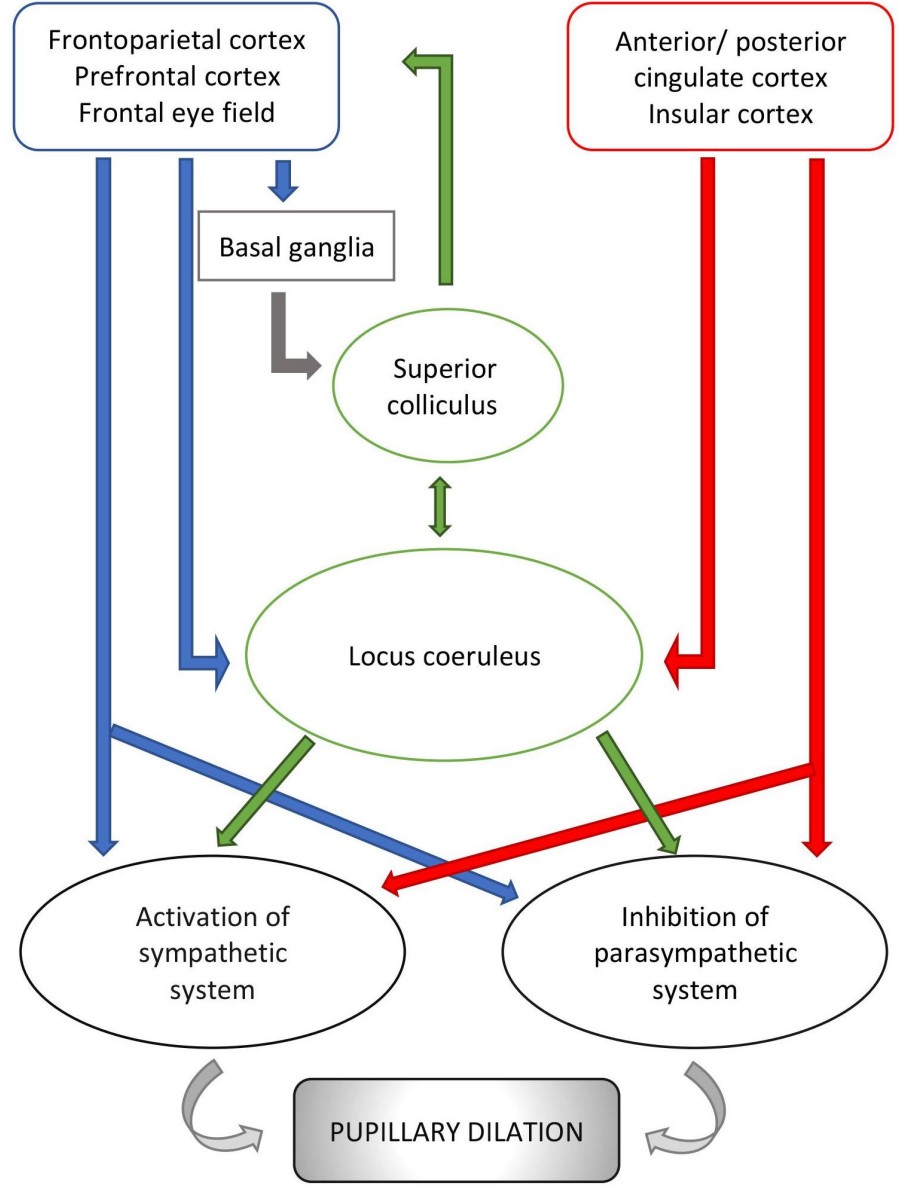

**Figure 2 Schematic representation of pupillary pathways that are activated during cognitive and emotional processes, including arousal and vigilance.** Pathways, connecting the cortical areas to the parasympathetic system and the sympathetic system, are inhibitory or activating. Neurons emerging from the locus coeruleus inhibit the parasympathetic system at the Edinger Westphal nucleus and activate the sympathetic system via connection to the spinal cord tract of the sympathetic system. Red arrows: connections from cortical areas involved in the autonomic control i.e., anterior/posterior cingulate cortex and insular cortex. Blue arrows: connections from other cortical areas involved in visual processes. Green arrows: connections from subcortical structures i.e., locus coeruleus and superior colliculus. For reference to Brodmann areas, see text.

even suppress, the activity of neurons with different tuning (*Schlag, Dassonville & Schlag-Rey, 1998*). From these observations, *Becket Ebitz & Moore (2017)* hypothesized that the frontal eye field and parts of the pretectum interact in regulating pupillary function.

Although evaluation of the pupils is part of the routine clinical examination, only few human studies have correlated pupillary function with specific cerebral areas to replicate results from (invasive) non-human primate studies. Systematic studies on pupil diameter have been conducted in three clinical settings: Raised intracranial pressure, which may lead to oculomotor nerve compression and brain herniation; traumatic brain injury and cerebrovascular disease; but only studies on the latter have provided data on candidate cerebral areas that may regulate the pupillary light reflex. The classical work on this topic is by *Herman (1975)*. In a retrospective study of 363 cerebral infarction patients, having excluded previous ocular pathology, local trauma, and active blood serology, the author reported that 5% of the examined patients had an asymmetrical pupillary response. Among the patients with pupillary asymmetry, 80% showed contralateral hemispheric stroke lesions, associated with other focal neurological signs and 20% of the patients had a dilated pupil homolaterally to the hemispheric lesion. A more recent work (*Peinkhofer et al., 2018*) found differences in the second phase of the pupillary light reflex, i.e., when pupils dilate back to baseline diameter, in patients with prefrontal eye field and/or insular infarcts (Brodmann areas 8, 13 and 16). In this study the authors assessed pupillary function in patients with an acute anterior circulation stroke, treated with endovascular thrombectomy, and compared patients with infarcts in the prefrontal eye field and/or insular cortex to patients with infarcts in other areas (based on neuroimaging). No difference was found in the overall pupillary function, but subtle changes were observed in the dilatation phase. Therefore, the prefrontal eye field and/or insular cortex may have a role in modulation of pupillary light reflex, influencing the autonomic system directly or indirectly, perhaps via connections to subcortical structures such as the locus coeruleus. Similarly, it seems that subjects with focal damage in ventral and medial prefrontal cortex (Brodmann areas 11 and 25) have a constant reduction of reward-induced autonomic pupil responses, compared to age-matched, healthy controls, confirming the involvement of these areas in the cortical modulation of pupillary light reflex (*Manohar et al., 2016*).

## Pupillary constriction

Pupillary constriction, induced by the parasympathetic system, is less frequently associated with cognitive or emotional processes than pupillary dilatation. It can be related to parasympathetic eliciting emotions such as disgust (*Ayzenberg, Hickey & Lourenco, 2018*) or memory tasks (*Naber et al., 2013*). The latter result is in contrast to the great majority of the studies on this topic (*Kafkas & Montaldi, 2015*; *Kafkas & Montaldi, 2012*; *Bradley & Lang, 2015*) that reveal pupillary dilatation but the difference seems to be mainly methodological, that is, related to the temporal evolution of the pupillary reflects analyzed: the first phase (i.e., constriction) or the second phase (i.e., dilatation), which are present in any task involving visual information processing.

Of course, pupillary constriction is mostly related to light stimuli. However, constriction following bright stimuli seems to go beyond the simple brainstem oculomotor

reflex (*Steinhauer, Condray & Kasparek, 2000*; *Henderson, Bradley & Lang, 2017*). For instance, subjects presented with half bright-half dark objects showed pupillary constriction when focusing on the bright side as opposed to dilatation when switching attention towards the darker side, suggesting that pupillary function depends more on the attended stimulus than on the amount of light (*Mathôt et al., 2014*; *Binda, Pereverzeva & Murray, 2013a*; *Binda, Pereverzeva & Murray, 2014*; *Binda & Murray, 2015*; *Mathôt et al., 2013*; *Mathôt et al., 2016*; *Naber, Alvarez & Nakayama, 2013*). Constriction was also observed with illusory images of brightness (*Laeng & Endestad, 2012*), with mental representation associated to light such as ''sunny skies'' (*Laeng & Sulutvedt, 2014*) and with written or spoken words such as ''lamp'', suggesting the presence of cortical influence of the brainstem light reflex pathway. Furthermore seeing intact images of the sun as opposed to scrambled images elicited a constriction (*Naber & Nakayama, 2013*; *Sperandio, Bond & Binda, 2018*; *Binda, Pereverzeva & Murray, 2013b*). Pupillary reaction influenced by images (*Laeng & Sulutvedt, 2014*), ambiguous stimulation (*Turi, Burr & Binda, 2018*) or memory tasks (*Blom et al., 2016*) has been suggested as tool to test subjective perception (*Turi, Burr & Binda, 2018*; *Laeng & Sulutvedt, 2014*; *Mathôt, Grainger & Strijkers, 2017*). In addition, experimental evidence for cortical control of the light reflex was provided by *Becket Ebitz & Moore (2017)*.

## Neuronal pathways

In light of these findings, the circuit behind pupillary function involves neuronal pathways connecting cortical regions to the locus coeruleus and the superior colliculus, two main pretectum structures. The locus coeruleus receives signals from the insular cortex (Brodmann areas 13 and 16), the anterior and posterior cingulate cortex (Brodmann areas 23, 24, 31, 32) and prefrontal cortex (Brodmann area 9, 11, 25). The superior colliculus receives inputs from frontal (Brodmann area 8), and frontoparietal cortex (Brodmann areas 6, 39, 40). Of note, Brodmann areas 6, 8, 39 and 40 might be connected to the locus coeruleus directly and indirectly via areas 13, 16, 23, 24, 31 and 32 (*Mill, O'Connor & Dobbins, 2016*; *Lehmann & Corneil, 2016*; *Wang et al., 2012*; *Joshi et al., 2016*; *Alnaes et al., 2014*). The locus coeruleus projects directly, and indirectly via the paragigantocellularis nucleus of the ventral medulla, to the Edinger-Westphal nucleus (*Joshi et al., 2016*). Similarly, the superior colliculus sends inputs directly, and indirectly via the mesencephalic cuneiform nucleus, to the Edinger-Westphal nucleus (*Wang et al., 2012*). However, there also exist pathways that connect the locus coeruleus with the superior colliculus directly (*Lehmann & Corneil, 2016*).

## Limitations

It should be noted that this systematic review has some limitations. First, we excluded studies with less than 15 patients, perhaps missing some relevant research. Second, the tools used to measure pupillary function were not the same across studies and, third, the exclusion criteria regarding previous neurological or ocular pathologies were not always clearly stated. Finally, it should be noted that pupillary function can be influenced by medication affecting the noradrenergic system, and very few papers provided information

about the presence of absence of such medication. On the positive side, this paper is the only recent review on the topic and includes more than 200 publications on cortical pathways and behaviors modulating pupillary function.

In summary, this review shows that:

- cognitive efforts and processing of emotional stimuli influence pupillary diameter in both humans and rhesus macaques, typically evoking pupillary dilatation;
- pupillary constriction occurs in response to light stimuli, both real and imagined, suggesting a cortical influence on subcortical reflex pathway;
- damage to salient cortical and subcortical areas such as frontal and prefrontal cortex, as well as key structures for autonomic control, seem to affect pupillary function by modulating the pupillary diameter;
- and micro stimulation of the frontal eye field (Brodmann area 8), locus coeruleus and superior colliculus in non-human primates leads to pupillary dilatation, suggesting involvement of these areas in the pupillary light reflex.

## CONCLUSIONS

Cognitive and emotional processes evoke a change in pupillary diameter, typically dilatation, in both humans and non-human primates, reflecting vigilance, arousal or attention. Stimuli related to light, whether real or imagined, elicit a pupillary constriction. Both dilatation and constriction are dependent on autonomic activation with cortical influence. The main structures involved are the locus coeruleus and the superior colliculus because of their direct and indirect connections to the Edinger-Westphal nucleus. Furthermore, cortical areas such as the prefrontal and the frontal cortex, particularly the frontal eye field (Brodmann area 8) and areas involved in autonomic control, such as insular cortex (Brodmann areas 13 and 16) and anterior cingulate cortex (Brodmann areas 24 and 32), modulate the pupillary light reflex via connections to subcortical structures and the Edinger-Westphal nucleus.

### Funding
The authors received no funding for this work.

### Competing Interests
The authors declare there are no competing interests.

### Author Contributions
- Costanza Peinkhofer performed the experiments, analyzed the data, contributed reagents/materials/analysis tools, prepared figures and/or tables, authored or reviewed drafts of the paper, approved the final draft.
- Gitte M. Knudsen authored or reviewed drafts of the paper, approved the final draft.

- Rita Moretti performed the experiments, analyzed the data, contributed reagents/-materials/analysis tools, authored or reviewed drafts of the paper, approved the final draft.
- Daniel Kondziella conceived and designed the experiments, performed the experiments, analyzed the data, contributed reagents/materials/analysis tools, prepared figures and/or tables, authored or reviewed drafts of the paper, approved the final draft.

## Data Availability

No raw data were generated; this is a systematic review.

## Supplemental Information

Supplemental information for this article can be found online at http://dx.doi.org/10.7717/peerj.6882#supplemental-information.

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
