# Peer review of "Cortical modulation of pupillary function: systematic review"

_PeerJ, doi:10.7717/peerj.6882_

## Round 0.1 · original submission · Major Revisions

Two reviewers have now read your manuscript and have made suggestions. Please address each suggestion in your revision and/or response letter.

Reviewer 1 ·

Basic reporting

After reading this manuscript I am looking for data collected,to be analysed and discussed using Broadmann Areas and white matter tracts especially that contribute directly and indirectly to pupillary function,
The responses differ in the growing cortex from neonates till the elderly brain cortex.This is missing in this review and makes it less informative and citable in the medical world.

Experimental design

The data collected from publications should include
Broadmann Areas and white matter tracts especially that contribute directly and indirectly to pupillary function,
The responses differ in the growing cortex from neonates till the elderly brain cortex

Validity of the findings

The systemic review should seek these data sets and include them(Broadmann Areas and white matter tracts especially that contribute directly and indirectly to pupillary function.
These responses differ in the growing cortex from neonates till the elderly brain cortex)

Reviewer 2 ·

Basic reporting

I enjoyed reading this brief and informative review on cortically driven pupil size changes.

The manuscript is adequately structured, it is written in a clear and coincise way and it covers a significant portion of the relevant literature. The authors set a very ambitious goal, a comprehensive review of the literature on cortical modulations of pupil size. The work goes a long way towards achieving this goal, although I see three areas that could be explored further.

1) there is evidence that attention can enhance the pupillary constriction in response to light - when the light stimulus is at the focus of attention. This effect is qualitatively different from the pupil dilation related to cognitive effort or to “paying more attention”: with a constant level of attention, directing its focus to a more luminous location or feature leads to pupil constriction, while focusing on a darker stimulus leads to pupil dilation (Binda et al., JNeuroscience 2013; JNeurophysiology 2014; JoV 2015; Sci rep 2017; Turi et al., eLife 2018; Mathot et al., 2013; 2014; 2016; 2017; Naber et al., 2013). This body of work has recently been supported by experiments in non-human primates, where stimulation of FEF produces the same pupillary effects as attention shifts: enhanced pupillary light response when the light stimulus is in the stimulated FEF receptive field (Ebitz and Moore 2017).
Thus, dilation is not the only effect of cortical signals on the pupil - although undoubtedly it is the most studied effect in the literature.

2) Naber and Nakayama 2013 is presented as an exception in the literature, reporting pupillary constriction of cortical origin. It is important to state that this work is not isolated. The very same result was reported by Binda et al. 2013 (same issue) using a different approach, and later replicated (e.g. Sperandio et al., 2018). Also, and perhaps most crucially, Laeng & Endestad 2012 had previously shown that pupil constriction can be elicited by illusory luminance increments.

3) There is growing interest in using pupillometry to track the contents of perception during ambiguous stimualtion (Einhäuser et al., 2008; Naber et al., 2012; Turi et al., 2018) and during imagery (Mathot et al., 2017; Laeng & Sulutvedt 2013) or working memory (Blom et al., 2016), both clearly involving cortical signals.

The studies in my #1, #2 and #3 indicate that turning attention towards a a brighter stimulus (be it a real external stimulus or an internal representation) leads to pupillary constriction. Thus, I do not think it is correct to say that cortical modulations are generally seen as a pupil dilation. There is a group of studies, using a variety of approaches to show that cortical signals can also interact with the pupillary light response and produce pupil constriction. I strongly recommend including these papers, as I feel the review would be more balanced and more useful with these.

Experimental design

The literature search is performed with clearly defined and systematic criteria. My only concern regards the exclusion of any study with less than 15 participants. Of course large Ns are good but why 15? I find it difficult to come up with a justification for picking a specific number (which ends up being different for humans and non human primates, by the way).

Validity of the findings

The conclusions from the literature review are well stated and linked to the original research question. However, I suggest that they are revised after including the three areas of research that I described above (field: Basic Reporting)

---

## Round 0.2 · accepted · Accept

One reviewer has concluded that you have addressed all concerns and that your manuscript is publishable.

# Reviewer 2 ·

Basic reporting

no comment

Experimental design

no comment

Validity of the findings

no comment

Additional comments

the authors have addressed all my comments